# Near-infrared spectroscopy outperforms genomics for predicting sugarcane feedstock quality traits

Mateus Teles Vital Gonçalves[1], Gota Morota[2], Paulo Mafra de Almeida Costa[3], Pedro Marcus Pereira Vidigal[4], Marcio Henrique Pereira Barbosa[5], Luiz Alexandre Peternelli[1] *

1 Departamento de Estatística, Universidade Federal de Viçosa, Viçosa, MG, Brazil, 2 Department of Animal and Poultry Sciences, Virginia Polytechnic Institute and State University, Blacksburg, VA, United States of America, 3 Instituto Federal Catarinense—Campus Concórdia, Concórdia, SC, Brazil, 4 Centro de Análises de Biomoléculas/NuBioMol, Universidade Federal de Viçosa, Viçosa, MG, Brazil, 5 Departamento de Fitotecnia, Universidade Federal de Viçosa, Viçosa, MG, Brazil

* peternelli@ufv.br

**Data Availability Statement:** The data underlying the results presented in the study are available from 10.6084/m9.figshare.12635717.

## Abstract

The main objectives of this study were to evaluate the prediction performance of genomic and near-infrared spectroscopy (NIR) data and whether the integration of genomic and NIR predictor variables can increase the prediction accuracy of two feedstock quality traits (fiber and sucrose content) in a sugarcane population (*Saccharum* spp.). The following three modeling strategies were compared: M1 (genome-based prediction), M2 (NIR-based prediction), and M3 (integration of genomics and NIR wavenumbers). Data were collected from a commercial population comprised of three hundred and eighty-five individuals, genotyped for single nucleotide polymorphisms and screened using NIR spectroscopy. We compared partial least squares (PLS) and BayesB regression methods to estimate marker and wavenumber effects. In order to assess model performance, we employed random sub-sampling cross-validation to calculate the mean Pearson correlation coefficient between observed and predicted values. Our results showed that models fitted using BayesB were more predictive than PLS models. We found that NIR (M2) provided the highest prediction accuracy, whereas genomics (M1) presented the lowest predictive ability, regardless of the measured traits and regression methods used. The integration of predictors derived from NIR spectroscopy and genomics into a single model (M3) did not significantly improve the prediction accuracy for the two traits evaluated. These findings suggest that NIR-based prediction can be an effective strategy for predicting the genetic merit of sugarcane clones.

## Introduction

The strides achieved with improved instruments, laboratory techniques, and bioinformatics tools have allowed the emergence of next-generation sequencing technologies [1]. These technologies can deliver DNA-level information at an ever more cost-effective and high-

**Funding:** MTVG received a masters degree scholarship (154611/2017-4) from the Conselho Nacional de Desenvolvimento Científico e Tecnológico (CNPq). This work was supported by the Coordenação de Aperfeiçoamento de Pessoal de Nível Superior - Brasil (CAPES) - Finance Code 001, the Fundação de Amparo à Pesquisa do Estado de Minas Gerais (FAPEMIG), and the Conselho Nacional de Desenvolvimento Científico e Tecnológico (CNPq) - Grant Number 310503/2015-9 to LAP. We are also thankful for the Inter-University Network for the Development of Sugarcane Industry (RIDESA) for all the field experiment support. The funders had no role in study design, data collection and analysis, decision to publish, or preparation of the manuscript.

**Competing interests:** The authors have declared that no competing interests exist.

throughput manner, which has boosted the important role genomic prediction (GP) might play in plant breeding [2]. The idea of GP is to fit a regression model using phenotypic records and the entire set of molecular markers concurrently. The model developed enables the prediction of the genetic merit of genotyped but non-phenotyped populations [3].

The application of GP as a breeding strategy is envisioned to reduce costs while saving time and resources [4]. One example is the reduction of generation interval as genitors could be crossed, the resulting progeny have their DNA collected from seeds or juvenile tissues, and then genotyped to have their breeding values predicted, what may result in gains per unit of time [5]. Sugarcane breeding programs are a good case point, which are characterized by long breeding cycles of phenotypic selection, performed over years of evaluation trials across different environments [6]. Another advantage of GP is that it could eventually lead to a reduction in phenotyping costs, especially in situations where traits are difficult to measure and when there are thousands of candidate genotypes to evaluate [7]. Therefore, the adoption of GP over phenotypic selection is expected to augment selection efficiency and to accelerate cultivar release [7,8].

Nevertheless, the implementation of GP is highly dependent on accurate phenotyping records [9]. Moreover, conventional phenotyping is not excluded from GP-based plant breeding schemes as model updates on new training populations would still be necessary [10]. However, besides the forecasts of continuous advances and decreasing costs in molecular breeding, the phenotyping step remains a significant bottleneck [11]. Phenotyping routine traits at breeding stations are commonly performed manually over several crop years and across different environments, which is often a time-consuming, labor-intensive, and high-cost task [12]. Also, conventional phenotyping is prone to human error. Thus, the genetic potential of populations may not be fully exploited [11].

Research efforts to address these constraints are currently being conducted with the development of high-throughput phenotyping (HTP) systems. HTP is an incipient, though a growing area of interest among plant breeders [13]. These novel approaches include a multitude of sensors and imaging techniques mounted on ground-based or uncrewd aerial vehicles (UAV) that can collect phenotypes in a precise, automated, and large-scale fashion [14]. Thus, HTP tools have the potential to improve plant breeding program pipelines significantly [15]. Moreover, HTP technologies can replace standard less effective phenotyping protocols, thus saving much time and resources. For instance, near-infrared (NIR) spectroscopy technology has been successfully applied for many crops, including sugarcane, to screen biological sample compositions and also for breeding purposes [9,16–18]. The attractive features of NIR spectroscopy could potentially aid sugarcane breeders, with the increase of selection accuracy and reduction of costs when compared to conventional phenotyping [11].

The combination of HTP systems with improved genomic tools is heralded to increase genetic gains in plant breeding [11,12]. However, the startling amount of data being generated is outpacing our ability to explore it. Besides, how this information can properly be implemented is still unclear and needs further investigations [14,19]. The integration of HTP information and GP can be performed by the exploitation of HTP platforms to provide phenotypic records that can be either treated as secondary traits (e.g., vegetation indexes) and regressed on molecular markers, or as predictor variables together with molecular markers in a single- or multi-trait analysis [20]. For instance, Crain et al. (2018) [21] investigated different proposals to integrate HTP derived variables into GP models in wheat and found improved prediction accuracies. Other strategies provide modeling alternatives that include interaction effects [22,23].

The main goal of this study was to investigate the performance of the integration of HTP and genomic datasets aiming to increase the accuracy of prediction for two important

sugarcane feedstock quality traits, namely fiber (FIB) and sucrose (PC) content, in a commercial sugarcane (*Saccharum* spp.) population from the sugarcane genetic breeding program of the Universidade Federal de Viçosa (PMGCA-UFV). The population was genotyped for single nucleotide polymorphisms (SNPs) and screened using NIR spectroscopy. We compared three modelling strategies: 1) genome-based prediction, 2) NIR-based prediction, and 3) the integration of SNP markers and NIR wavenumber variables as predictors.

## Material and methods

### Plant material

In this study, we evaluated a population of 385 clones derived from an originally seedling population of 98 half-sib families. The seedling population, in which each plant is a single genotype, was the result of crosses made at the Serra do Ouro Flowering and Breeding Station, municipality of Murici, Alagoas State, Brazil (09˚13' S, 35˚50' W, 450 m altitude). After processing, seeds were sent to the Sugarcane Genetic Breeding Research Station (CECA) of the Universidade Federal de Viçosa, municipality of Oratórios, Minas Gerais State, Brazil (20˚25' S, 42˚48' W, 494 m altitude) and germinated in a nursery house. Subsequently, seedlings obtained from each family were transplanted to the field and evaluated in first (plant cane) and second (ratoon) crops based on desirable traits in first (T1) and second (T2) clonal trial stages of selection [6].

### Experimental design

An augmented block design was initiated in May 2016 at the CECA municipality of Oratórios, Minas Gerais State, Brazil (20˚25' S, 42˚48' W, 494 m altitude). The released cultivars RB867515, RB966928, and RB92579 were included as checks once in each block, and regular unreplicated clones were arranged in 21 blocks [24]. The replicated checks are well-established cultivars, widely grown throughout the country and are often used as parents for crosses. The experimental plots consisted of double-row 3 m long furrows × 1.4 m between rows and clones were cultivated following standard agronomic protocols regarding fertilization, weed control, and pest management [25]. Buffer rows of released cultivars encompassed the whole experiment area.

### Phenotypic data

The clones were evaluated in the first ratoon (second crop) 26 months after planting. The method employed to estimate the percentage of FIB content followed recommendations of the CONSECANA manual [26]. Sugarcane breeding programs in Brazil routinely apply these protocols. Harvest and quality analyses were performed in July 2018. To obtain a representative set of the samples, ten randomly selected stalks from each of the double-row plots were cut at ground level with a machete. Green tops, clinging leaves, and leaf sheaths were removed before stalks were bundled and weighted using a dynamometer (S1 Fig). These ten randomly selected stalks in each plot were shredded. A subsample of 500 g from the shredded stalks was collected and pressed with a hydraulic press. After pressing, the remainder fiber cake was collected and taken to the laboratory. We obtained the BRIX% (percentage of soluble solids) and POL% (percentage of sucrose) values from the juice. The BRIX was obtained with a refractometer (HI96801 Model, Hanna® instruments, Woonsocket, USA), while POL was obtained by polarimetry using a saccharimeter (SDA2500 Model, Acatec, Brazil) after clarifying the solution with lead acetate. The remainder fiber cake was weighed (*WC*) and used to derive fiber

content [27]:

$$FIB = 0.08 \times WC + 0.876 \tag{1}$$

The apparent percentage of sucrose in sugarcane (PC%) was derived based on POL% and FIB as follows:

$$PC = POL\% \times (1 - 0.01 \times FIB) \times C \tag{2}$$

where *C* is the coefficient to convert sucrose of juice into sucrose of cane calculated using the formula *C = 1.0313–0.00575 × FIB*. The final values were all expressed as the total fresh biomass basis (500 g of shredded stalks).

## Sample preparation and NIR spectra acquisition

Another subsample of 100 g from the shredded stalks was collected and immediately taken to dry in a forced-air circulating oven at 50˚C for 24 h or until a constant mass was reached (S1 Fig). Dried samples were then ground, packaged in a plastic zip bag, and stored. The NIR spectra of samples were measured in indoor-conditions at room temperature of 21˚C. The instrument used was a Fourier transform near-infrared (FT-NIR) spectrometer set (Antaris™ II Model, Thermo Scientific Inc., USA), under the following operating conditions: 4 cm$^{-1}$ resolution in an investigated wavenumber range of 10000 to 4000 cm$^{-1}$ and reflectance mode as log (1/R), where R is the measured reflectance. Samples were placed into a powder sampling cup accessory and arranged into the instrument window. At each scan, the accessory was moved to cover different positions of the sample, totaling six positions. A single scan measure was the average result of 32 scans. For each sample, a total of 192 scans were made and then averaged, representing the final spectrum. The final NIR matrix used in the subsequent analyses had a dimension of 385 rows and 3,112 columns.

## DNA isolation, sequencing, and genotyping data

Sugarcane DNA samples were isolated using DNeasy Plant Mini Kit (QIAGEN, Hilden, Germany) and sent to RAPiD Genomics (Gainesville, Florida, USA) for the construction of probes, sequencing, and identification of molecular markers. Samples were genotyped using single-dose SNP markers based on the Capture-Seq technology (https://www.rapid-genomics.com). Raw sequence reads were mapped, called, and filtered. Reads were anchored to a monoploid reference genome of sugarcane (*Saccharum spp.*) [28] using the BWA-MEM algorithm of the BWA version 0.7.17 [29]. A flag identifying the respective sugarcane genotype was added to each mapping file. The mapping files were processed using SortSam, MarkDuplicates, and BuildBamIndex tools of Picard version 2.18.27 (https://github.com/broadinstitute/picard/). Variants were called using FreeBayes version 1.2.0 (https://github.com/ekg/freebayes) with a minimum mapping quality of 20 (probability of miscalling), minimum base quality of 20 (SNPs with missing data higher than 20% were eliminated), and minimum coverage (how many times a fragment was sequenced) of 20 reads at every position in the reference genome. Thereafter, the SNP marker matrix was coded counting the occurrence of the reference allele A. Thus, considering the genotypes AA, Aa, and aa, the matrix entries would be 2 (homozygosity for the reference allele), 1 (heterozygosity with one reference and one alternative allele), and 0 (homozygosity for the alternative allele), respectively. Further, markers with minor allele frequency lower than 5% were eliminated. Lastly, missing variants were imputed from a binomial distribution density function using the frequency of the non-missing variants. A total of 124,307 SNPs was retained for further analyses.

## Statistical analysis

A two-stage analysis was employed. In the first step, we run a mixed model equation with variance components estimated by REML using the SELEGEN-REML/BLUP software [30]. We considered the model

$$\boldsymbol{y} = \boldsymbol{1}\mu + \boldsymbol{Xb} + \boldsymbol{Zg} + \boldsymbol{\varepsilon} \tag{3}$$

where **y** is the vector of phenotypes; **1** is a vector of 1s; μ is the overall mean; $\mathbf{b} \sim N(\mathbf{0}, \mathbf{I}\sigma^2_b)$ is the vector of random block effects; $\boldsymbol{g} \sim N(\mathbf{0}, \mathbf{I}\sigma^2_g)$ is the vector of random genetic effects, and $\boldsymbol{\varepsilon} \sim N(\mathbf{0}, \mathbf{I}\sigma^2_e)$ is the vector of residuals. The effects of checks were taken as random. The capital letters **X** and **Z** represent the incidence matrices of the respective random effects. In the second step, the best linear unbiased predictors (BLUPs) of each trait were used as dependent variables using three prediction models (M1, M2, and M3) to evaluate the prediction accuracy for FIB and PC.

The first (M1) was a single-trait model using only markers which takes the following form:

$$\boldsymbol{y}^* = \boldsymbol{1}\mu + \boldsymbol{Wa} + \boldsymbol{\varepsilon} \tag{4}$$

where $\boldsymbol{y}^*$ is the vector of adjusted phenotypic values (BLUPs) for FIB or PC, **1** is a vector of 1s; μ is the overall mean, **W** is the matrix with SNP markers for each individual, $\boldsymbol{a}$ is the corresponding vector of marker effects, and $\boldsymbol{\varepsilon}$ is the vector of residuals.

The second (M2) was a single-trait model using NIR wavenumber variables as predictors:

$$\boldsymbol{y}^* = \boldsymbol{1}\mu + \boldsymbol{Ns} + \boldsymbol{\varepsilon} \tag{5}$$

where $\boldsymbol{N}$ is the matrix with the spectrum for each individual along the wavelength, and $\boldsymbol{s}$ is the corresponding vector of wavelength effects. It is a commonplace to apply mathematical transformations to the NIR matrix before analysis to increase the signal to noise ratio [31]. More details can be found elsewhere [32]. We tested different combinations of pre-processing techniques. The pre-processing combination that yielded the best results was Savitzky-Golay smoothing (SGS) (window: 5; polynomial order: 2) followed by multiplicative scatter correction (MSC) and mean centering (MC) for FIB, whereas for PC the best combination was SGS (window: 5; polynomial order: 2) and MC (Fig 1).

In the third model (M3), we combined SNP markers and NIR wavenumber variables as predictors fitting the following linear model:

$$\boldsymbol{y}^* = \boldsymbol{1}\mu + \boldsymbol{Wa} + \boldsymbol{Ns} + \boldsymbol{\varepsilon} \tag{6}$$

In Eq (6), the pre-processing combination of the NIR matrix that best contributed with the SNP matrix for maximizing the prediction accuracy were SGS (window: 5; polynomial order: 2), MSC and MC for FIB, and SGS (window: 5; polynomial order: 2), 1° derivative, MSC and MC for PC. The incidence matrices $\boldsymbol{N}$ and $\boldsymbol{W}$ were scaled (centered and standardized) in all prediction models prior to the analyses.

## Regression models

The models were tested using two regression methods: The BayesB and partial least squares (PLS). BayesB is a hierarchical Bayesian approach that performs variable selection and shrinkage [3]. We used a multi-layer BayesB by assigning different independent priors for SNP markers and NIR wavenumbers in M3. PLS regression is a dimension reduction method and fundamentally transforms the original collinear predictors into non-correlated variables [33]. In PLS regression, the algorithm identifies the principal components (latent variables) that

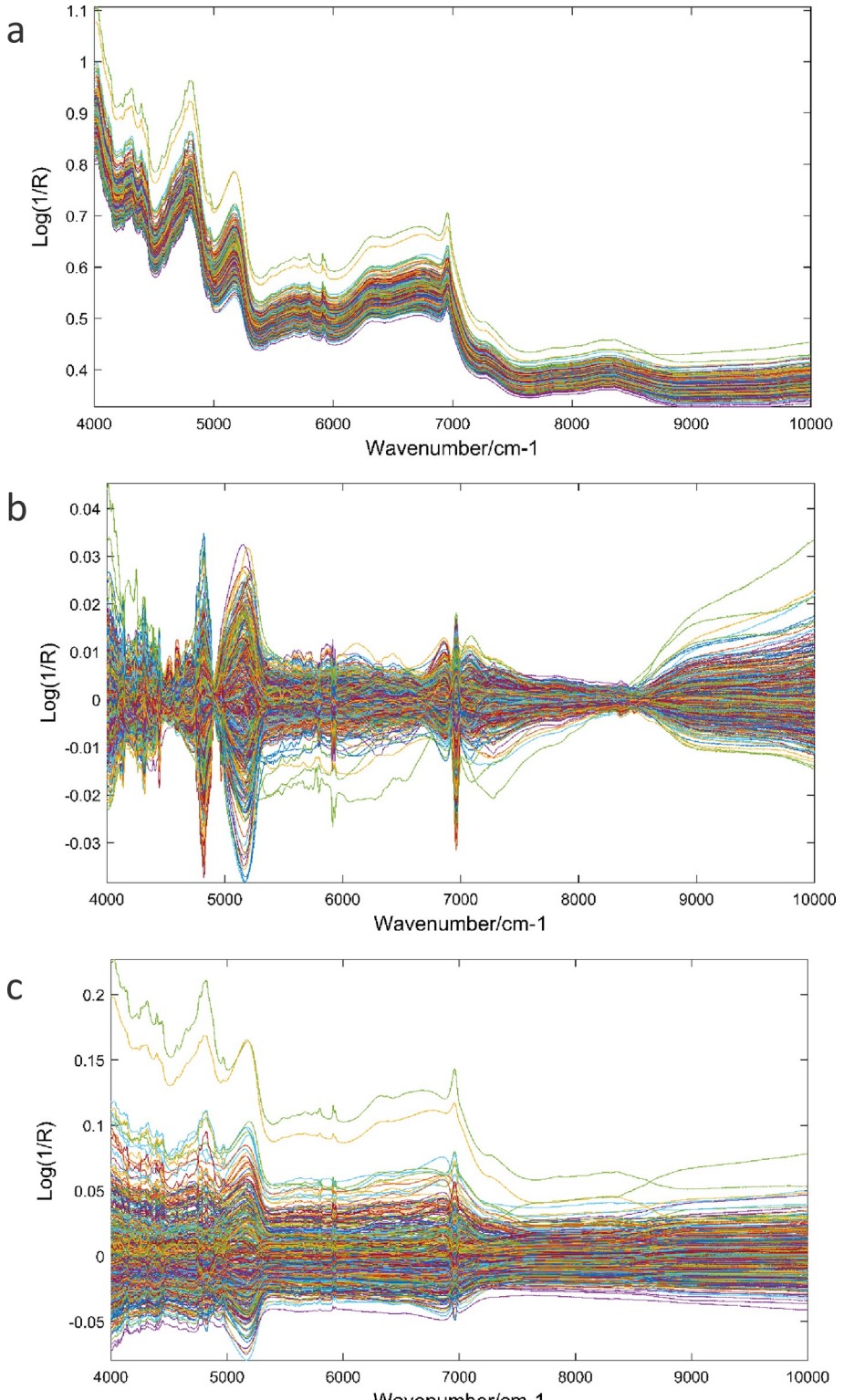

**Fig 1.** Sugarcane samples raw spectra (a) and pre-processed spectra for fiber content (b), and sucrose content (c).

best describe the data in terms of variance, and it does so by constructing linear combinations of all predictors. Furthermore, unlike other dimension reduction models such as principal component regression, the fitting procedure of PLS involves finding the latent variables that maximize the covariance between the predictors and phenotypes while minimizing the error [34,35].

The BayesB analyses were carried out using the BGLR package [36]. We run the BayesB for 25,000 samples, with the first 10,000 being discarded (burn-in) with a thinning interval of 10. The PLS regression was performed using the mixOmics package [37]. Both methods were implemented in R [38].

## Accuracy of predictions

The prediction accuracy of models M1, M2, and M3 was evaluated by random sub-sampling cross-validation repeated 20 times. The models are fitted using the data of the training set observations and tested to predict unknown samples of the validation set. In this study, the training set contained 80% of the samples (308 clones) and the validation set included the remainder of 20% (77 clones). At each time, the algorithm randomly selected a different subset of observations assigned to the training and validation sets. The results were compared by computing the mean Pearson correlation coefficient between observed and predicted values. A schematic diagram summarizing the whole experimental preparation and processing is depicted in Fig 2.

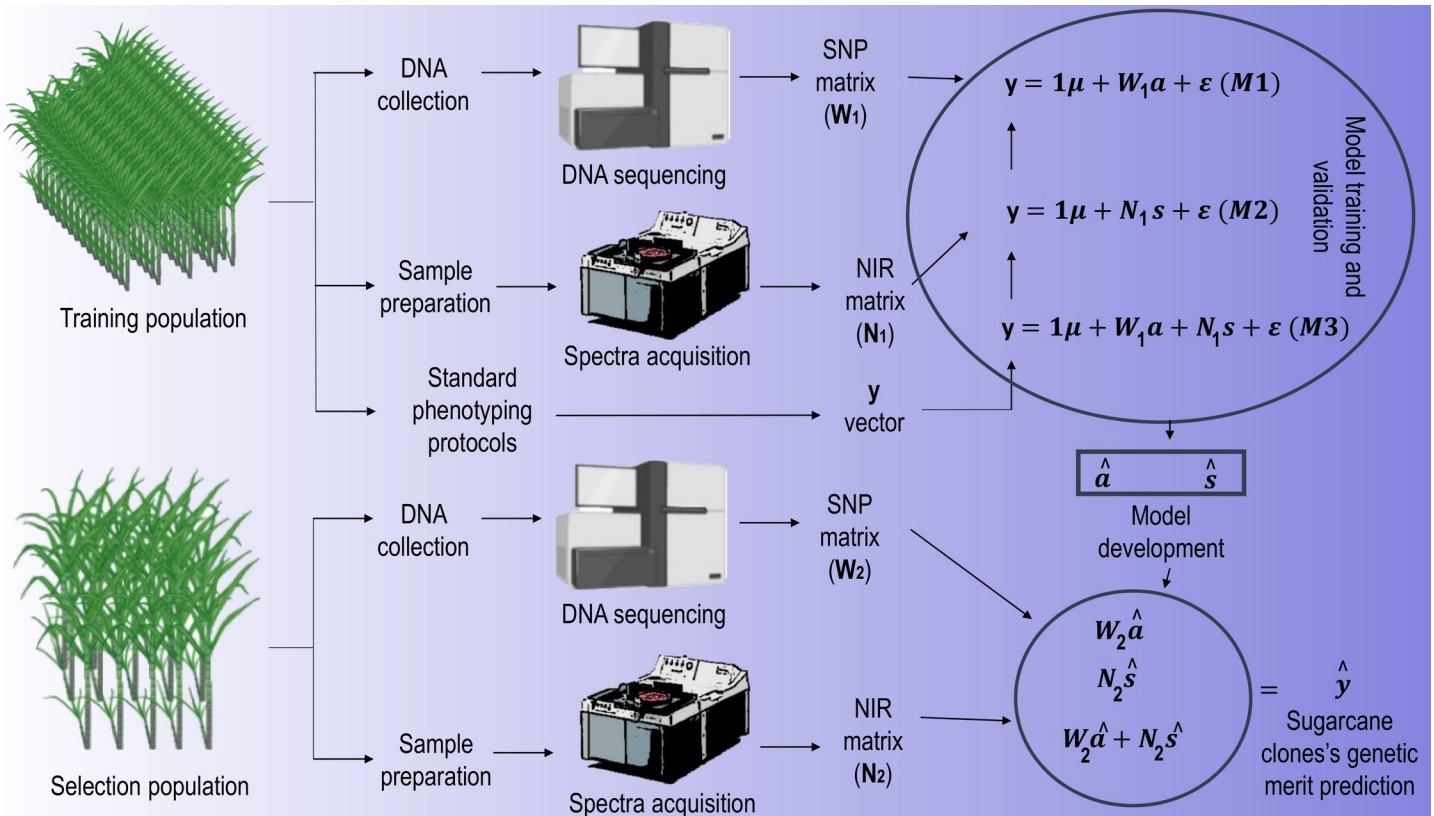

**Fig 2. Schematic diagram of the experimental procedure.** Created with Biorender.com.

## Results

### Phenotypic data

We fitted a bivariate GBLUP (not shown) to estimate genetic correlation between the two traits; however, we found similar results compared to the correlation using univariate BLUPs. Since there was no difference, we decided to show the correlation using BLUPs. Fig 3 shows the correlation analysis using the BLUPs of FIB and PC. The data followed a Gaussian distribution curve. The two traits evaluated are negatively correlated (r = -0.22; p < 0.001).

The variance component estimates calculated using the REML/BLUP procedure were used to derive genetic and environmental parameters. Significant values (p < 0.01) of genetic variance ($\hat{\sigma}_g^2$) were observed from the deviance analysis for FIB and PC (Table 1). The estimates of individual broad-sense heritability ($h^2$) for FIB were considerably high. In contrast, the $h^2$ for PC was low. The heritability values obtained might have been the result of environment variance and the choice of the experimental design.

### Prediction models

The BayesB results from cross-validation are presented in Fig 4. We found that M1 resulted in the lowest prediction accuracy for FIB and PC. Additionally, M1 models showed the largest cross-validation uncertainty. The highest prediction accuracy of M2 was obtained for FIB (0.6138), followed by PC (0.5447). The combination of SNP markers and NIR spectra in M3 models yielded an increase in the predictive ability for PC (0.5860) and a marginal improvement for FIB (0.6231) in comparison to the models fitted using only NIR spectra (M2) as predictor variables. However, Tukey's test indicated no significant (*P* > 0.05) difference in predictive performance between M2 and M3 for both traits.

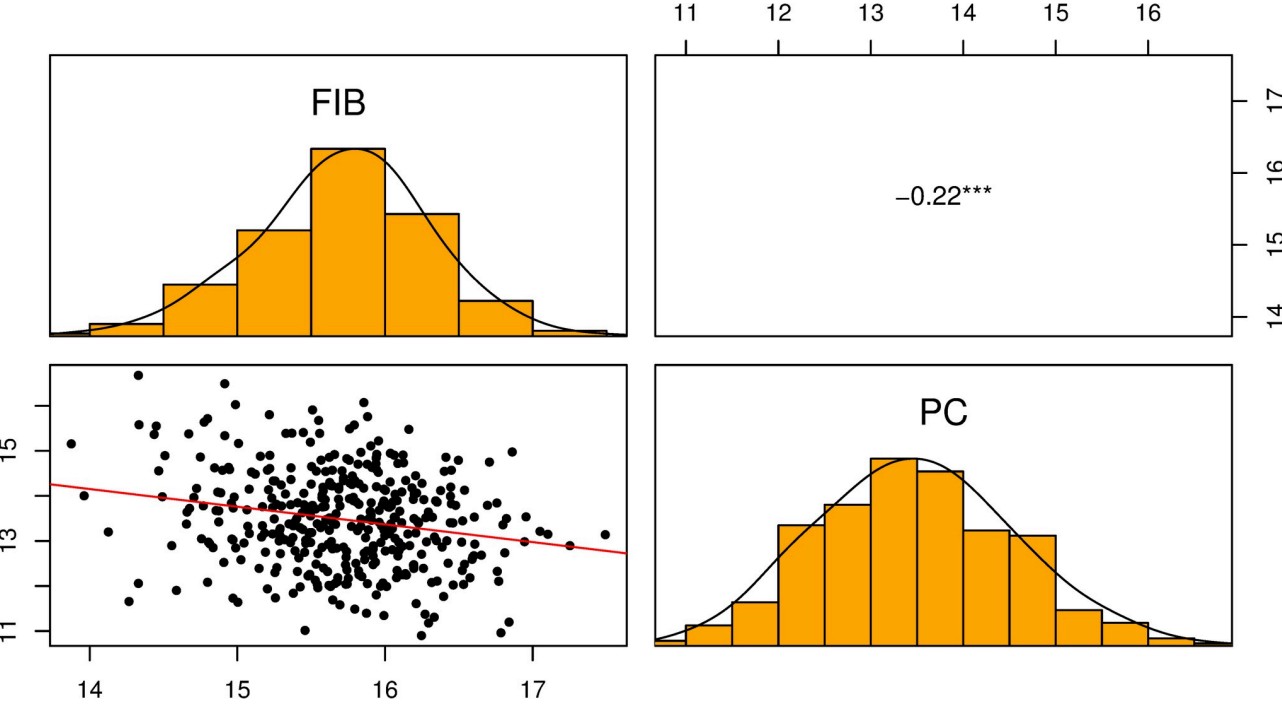

**Fig 3. Scatter plot, histogram, and correlation of BLUPs of FIB and PC for 385 sugarcane clones.**

**Table 1. Genetic and environmental parameter estimates of the 385 sugarcane clones evaluated.**

| Parameters | FIB% | PC% |
|---|---|---|
| $\hat{\sigma}_g^2$ | 1.3559* | 1.0727* |
| $\hat{\sigma}_e^2$ | 0.1861 | 2.3032 |
| $h^2$ | 0.8374 | 0.3176 |
| General mean | 13.46 | 15.70 |

$\hat{\sigma}_g^2$: Genetic variance effect; $h^2$: Individual plots broad sense heritability; FIB%: Fiber content; PC%: Sucrose content;
* significant at 1% probability by the analyses of deviance.

The cross-validation results using PLS regression are shown in Fig 5. Similar to BayesB, M1 models fitted using PLS resulted in the lowest prediction accuracies for both traits. The M2 model for FIB presented the highest prediction accuracy across prediction models (0.3917). Considering M3 models we observed a small increase, although no significant ($P > 0.05$), for PC (0.3942) compared to the M2 model (0.3673). In contrast, no improvement in prediction accuracy was observed for FIB.

## Discussion

In the present study, we explored different strategies to incorporate genomic and HTP derived information from NIR spectroscopy into a sugarcane genetic breeding program aiming to improve prediction accuracies. Jannink et al. (2010) [39] reported the similarity of NIR spectroscopy and GP approaches, as they inherently share the same purposes and statistical analysis challenges. For instance, the application of NIR spectroscopy aims to replace demanding and expensive laboratory protocols by developing statistical prediction models using high-

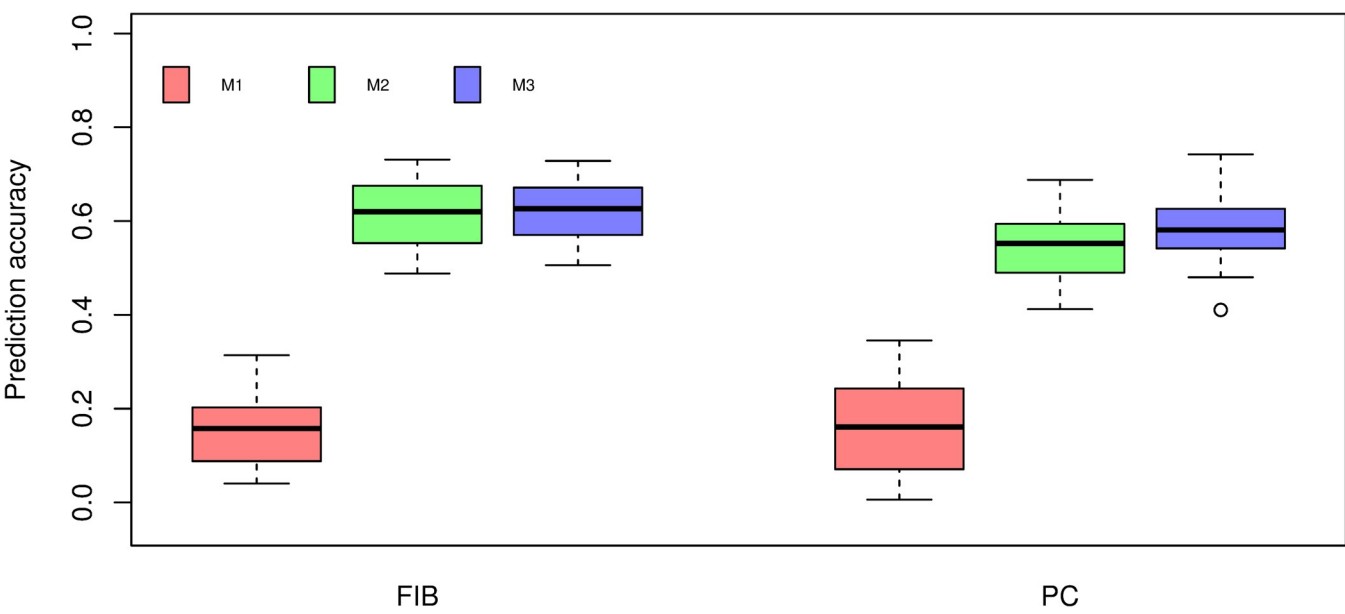

**Fig 4. Box-plot of cross-validation prediction accuracy of fiber content (FIB) and sucrose content (PC) using BayesB under three different prediction models.** M1: Markers. M2: Near-infrared spectra. M3: The combination of markers and near-infrared spectra. Different lowercase letters denote significant differences with Tukey's test ($P < 0.05$).

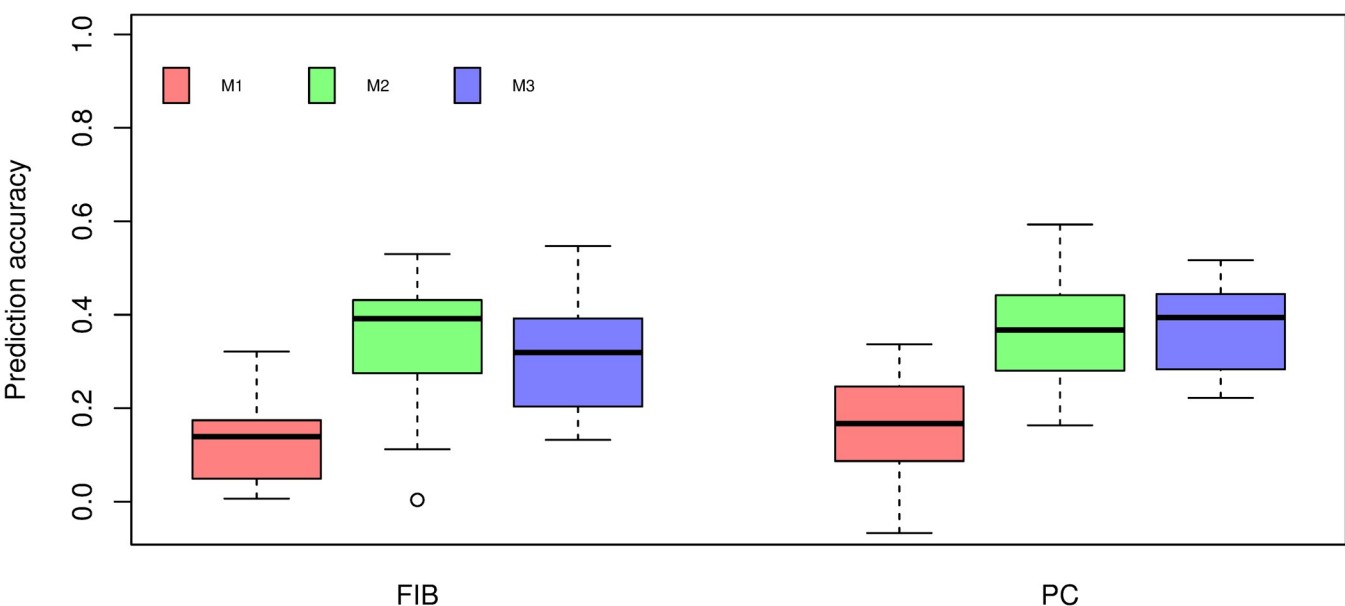

**Fig 5. Box-plot of cross-validation prediction accuracy of fiber content (FIB) and sucrose content (PC) using PLS under three different prediction models.** M1: Markers. M2: Near-infrared spectra. M3: The combination of markers and near-infrared spectra. Different lowercase letters denote significant differences with Tukey's test ($P < 0.05$).

dimensional input variables. Likewise, GP is intended to utilizes multivariate statistic methods to build prediction models that associate difficult to record plant phenotypes with easy to measure variables (e.g., SNPs) [40]. Moreover, the existing statistical methods capable of coping with the challenges associated with high-dimensional statistical analysis are used across research fields [21,41–44].

Along with the integration of genomic and spectroscopy datasets, we also considered their application as single predictors in the statistical models and evaluated the impact in the prediction accuracy of two sugarcane feedstock quality traits using a commercial sugarcane population of 385 individuals and tested two regression methods. We assessed the performance of PLS regression, which is arguably the most employed method when dealing with NIR datasets [45] and BayesB for the GP models. For all models, we employed repeated random sub-sampling cross-validation to assess the accuracy of prediction models.

## Phenotypic data

The results we found for genetic effects reveal the presence of genetic variance to be exploited for selection purposes. These results are in agreement with the study of Baffa et al. (2014) [46] and Wang et al. (2008) [47] for commercial sugarcane populations.

The estimated broad-sense heritability values indicate that the selection for superior clones based on phenotypic values would be effective for FIB, since environment variance showed little impact, but not for PC [48]. Ramos et al. (2017) [49] reported similar values of $h^2$ for FIB and PC.

The negative correlation observed between FIB and PC supports the dynamic of carbon partitioning and the antagonistic metabolic pathway of fiber and sucrose synthesis in sugarcane related in other studies, which suggest the trend of PC being negatively impacted by the increase of FIB [50,51].

## Prediction models

Overall, models fitted using BayesB were most predictive than PLS models for both traits, with accuracy estimates ranging from 0.5860 to 0.6231 (Fig 5). This result is in agreement with the study of Ferragina et al., (2015) [44], in which Bayesian models outperformed PLS regression for NIR-based prediction of dairy traits. Additionally, Solberg et al. (2009) [52] reported that the prediction accuracy of genome-wide breeding values from PLS regression was lower compared to that of BayesB.

The GP models (M1) showed poor predictive ability and only explained a small portion of the phenotypic variation of the two traits evaluated. The predictive ability of GP models observed was lower than that found by Gouy et al., (2013) [42]. The authors reported accuracies for within cross-validation panels considering bagasse content (an equivalent measure of FIB) and BRIX, which is highly correlated with PC [53], of up to 0.5 and 0.62, respectively. However, they used a DArT low-density molecular marker panel, whereas we used a high-density SNP panel. The extent of coverage is a critical aspect of GP [54,55]. However, Sousa et al. (2019) [56] applied GP to polyploid hybrids of *Coffea* spp. and reported that once the optimal number of SNPs is reached, a plateau is observed followed by a decrease in predictive accuracies. Further, Yang et al. (2017) [57] reported that for sugarcane and other crops with complex genomes, the quality of sequencing might be more important than a large number of SNPs suggesting that sequencing depth is crucial to filter low-quality sequence reads.

Deomano et al. (2020) [58] tested GP in sugarcane and obtained prediction accuracies of up to 0.45 for commercial cane sugar, a trait that is related to PC. By contrast, the highest prediction accuracy we observed in the present study for PC was of 0.1607. One possible explanation for this result might be the fact that the authors used a considerably larger training population than ours. It has been reported that increasing training population size can result in better predictive ability [59]. In the case of sugarcane and other polyploids, this factor is expected to play an even bigger role, given the expected allelic diversity [60–62].

Another factor that could explain the low level of prediction accuracies we observed is that the GP models fitted in this study only considered additive effects. Zeni Neto et al. (2013) [63] reported that additive and non-additive genetic effects are equally important for the determination of complex traits in sugarcane. Hence, the inclusion of non-additive effects in GP models for sugarcane and other clonally propagated species may improve prediction accuracies [64]. Results obtained by Denis et al. (2013) [65] in a simulation study, and de Almeida Filho et al. (2014) [66] using data from a full-sib population of loblolly pine (*Pinus taeda* L.) indicate the improvement of prediction accuracy when accounting for dominance effects. Nevertheless, according to Wei et al. (2016) [67], the slow rate of genetic gains reported for sugarcane yield in recent decades can be partly attributed to the low levels of narrow-sense heritability that most commercial sugarcane breeding populations exhibit [47,68,69]. However, the authors also stated that this could be a poor estimation of additive genetic component based on phenotypic evaluation. Therefore, the genomic models that account only for additive effects would still have usefulness to better estimate genetic values and help identify superior genitors.

In sugarcane and other polyploid crops, the estimation of allele dosage might improve predictions of GP models [70,71]. The utilization of single-dose marker systems in ployploids is considered less informative [72]. For instance, given the high heterozygosity and varying ploidy levels of sugarcane cultivars, for every cross each locus will segragate into multiple genotype classes. Hence, by using single-dose markers, different heterozygous genotypes will not be distinguished [73]. Recent studies have investigated polyploid parametrization and reported improvment in predicitons [71,74]. However, the big and complex poly-aneuploid genome of modern sugarcane hybrids has hindered the development of tools that provide a reliable

estimation of allele dosage [75]. These pieces of evidence suggest that significant bottlenecks in the application of GP to sugarcane are sequencing and sequencig data processing, as well as proper statistical methods to handle the complex inherent patterns of polyploidy [73,76]. Some research efforts to meet these limitations are underway [73,77,78].

The M3 modelling strategy we evaluated in this study aimed to increase accuracies by combining NIR wavenumbers and SNP markers to predict sugarcane clones' phenotypes for possible release as cultivars. This approach has been tested using different omic predictors (e.g., metabolomic and transcriptomic datasets [79–82]). Riedelsheimer et al. (2012) [80] jointly used metabolites and SNPs to predict general combining ability of maize hybrids. However, the authors found no improvement in predictions. Crain et al. (2018) [21] investigated a similar application we proposed herein and found the same trend. The authors evaluated the effect of including HTP data into GP models in different stressed environmental conditions. The results of their study revealed that models including HTP derived information as single predictors contained most of the predictive ability when compared to models with markers alone in one of the evaluated scenarios, which is consistent with the results we observed. We expected that the combination of M1 and M2 models would bring synergy and thus, improve model performance. However, the combination of predictors produced no significant improvement in comparison to the use of NIR wavenumber predictors alone (M2). Seemingly, most of the variation of the two evaluated traits that were captured by the M3 model came from the NIR spectra. Indeed, according to Rutkoski (2019) [83], the use of molecular markers is rather an indirect form of selection because it is performed based on genotypes. In contrast, selection using NIR spectroscopy is performed directly on phenotypes, which could be a possible reason to explain this result.

A wide range of references regarding the utilization of NIR spectroscopy in agriculture-related topics is available. For instance, studies employing NIR as an analytical tool include post-harvest quality monitoring [84], toxic compounds detection in seeds [35], and grain composition determination [85,86]. Further, Hayes et al. (2017) [87] performed NIR predictions of 19 wheat end-use quality traits using multi-trait analysis and obtained improved accuracies of genomic predictions. In the context of sugarcane, applications are focused on screening biomass sample physical properties and chemical composition [51,88–92]. Nevertheless, examples that make use of NIR spectroscopy for breeding purposes are also available and include the classification of sugarcane clones based on quality parameters [93], resistance to diseases [94], and pests [17]. Moreover, plant breeders have been benefiting from NIR spectroscopy using spectrometer sensors coupled with UAV and ground-based platforms [11,95,96]. Rincent et al., (2018) [97] proposed an approach in which relationship matrices are derived from NIR spectra data and compared the efficiency of predictions with standard GP models considering markers. Krause et al., (2019) [98] extended this concept by using hyperspectral reflectance derived relationship matrices and by modeling genotype × environment interactions. The results found by these authors suggest that models developed using NIR data can outperform GP models. Likewise, we observed with our dataset that NIR-based models alone provided better results. Finally, NIR spectroscopy may offer more opportunities to assist plant breeders with the advent of portable low-cost instruments [99,100].

## Genomic prediction and NIR spectroscopy implementation

Sugarcane is cultivated in a semi-perennial scheme and its multiplication is performed by vegetative propagation [101]. The initial step of an ordinary sugarcane breeding program consists of crossing pre-selected elite progenitors [6]. Modern cultivated sugarcane clones feature a complex genome structure, rendering each cross unpredictable [102]. Consequently, large

progeny populations are generated [101]. After crossings, the first stage of clonal selection at the PMGCA-UFV is referred to as T1 [103]. At T1, an array of limitations including physical space and short supply of propagation material precludes the installation of appropriate statistical experimental designs, which contributes to diminishing the selection accuracy, especially regarding low heritability traits [25]. Moreover, most sugarcane traits are believed to be quantitatively inherited [104,105]. Therefore, the subsequent stages of selection involve capital intensive field trials over multiple sites and years. Today, one breeding cycle of phenotypic selection in sugarcane can take up to 13 years [105,106].

The optimal strategy to incorporate GP into breeding schemes is not straightforward and needs consideration, since it can be influenced by many factors [107]. In the context of sugarcane breeding, some authors have argued that the main benefit of applying GP could come from the length reduction of breeding cycles by performing early selection of parental clones [58,105]. Sugarcane breeding programs typically adopt into their pipelines intrapopulational recurrent selection (IRS) schemes [108]. In IRS schemes, clones are reintroduced to the breeding cycle for new hybridizations as candidate superior parents. Presently, the strategy at the PMGCA-UFV is to evaluate candidates for selection in time-consuming field trials with the classical BLUP methodology, using phenotypes and pedigree records for the estimation of breeding values [109]. However, several studies indicate that the utilization of the realized relationship matrix based on markers is preferred because it allows the estimation of the Mendelian sampling term and is less prone to errors [110,111]. Moreover, the combination of genomic and pedigree information could provide a more reliable estimation of breeding values for parental selection in the IRS scheme, thereby improving selection accuracy [6,58,112]. Further, even with low accuracies, gains per unit of cost can be obtained when compared to conventional phenotypic selection with the elimination of seedlings that exhibit poor performance, i.e., lowest genomic estimated breeding values before the installation of early field trials; therefore, saving resources and optimizing the next selection stages [112–114]. Nevertheless, costs need to be thoroughly considered before the routine implementation of GP in this stage because a massive number of seedlings are generated to form the initial base populations.

According to Yadav et al., (2020) [105], the costs associated with high-throughput genotyping can be a major impediment for the adoption of large scale GP in sugarcane breeding. However, the decreasing costs of genotyping platforms and the integration with HTP platforms is encouraging [1,11,40]. The combination of HTP platforms (e.g., NIR spectrcopy) and GP is heralded to promote genetic gains by improving selection accuracy and reducing costs, mainly related to conventional phenotyping sessions [2,11]. From this perspective, NIR spectroscopy could be most useful to train GP models, screening more efficiently large training populations in order to estimate and continuously re-estimate marker effects [10]. In this setting, phenotypes are the predictions based on the NIR spectra. Alternatively, NIR data finds applicability in indirect selection, with the inclusion of phenotypes based on NIR spectra or spectra derived indexes into multi-trait genomic prediction models [87,115]. Further, NIR spectroscopy could likely be used to select superior clones for cultivar development with the replacement of costly and less effective phenotyping protocols in advanced selection stages thus, maximizing resources. The trend of decreasing costs in NIR instrumentation is clear, with the advent of portable instruments [99]. Therefore, it seems likely to be readily available for implementation as a low-cost phenotyping tool.

## Conclusion

Our experimental results showed that GP models had the lowest prediction accuracies. In addition, the combination of NIR wavenumbers and SNP markers as predictor variables did

not demonstrate significant improvements in accuracy to predict FIB and PC, when compared to the models solely based on NIR wavenumbers. NIR wavenumber predictors alone achieved high prediction accuracies for the two traits assessed in this study, indicating the potential usefulness of NIR spectroscopy to train GP models and for predicting total phenotypic value of sugarcane clones for possible release as cultivars. We speculate that the combination of GP and NIR spectroscopy has the potential to enable genetic gains and accelerate the release of new cultivars in sugarcane breeding programs by reducing the length of breeding cycles and improving selection efficiency.

## Supporting information

**S1 Fig. Overview of data acquisition.** A: stalks being harvested from double-row plots; B: stationary forage chopper machine used to shred stalks; C: hydraulic press used to extract the fiber cake and juice samples; D: fiber cake being weighted; E: samples being dried at a forced-air circulating oven; F: dried ground samples placed onto the NIR instrument window; G: saccharimeter instrument; H: sample spectrum displayed on the computer screen.
(TIF)

## Acknowledgments

The authors thank Professor Dr. Luis Antônio dos Santos Dias for providing the NIR instrument used in this study. Also, we acknowledge Professor Dr. Reinaldo Francisco Teófilo and Dr. Jussara Valente Roque for the helpful assistance regarding NIR spectra collection and downstream analyses. Finally, we acknowledge the numerous co-operators from the Sugarcane Genetic Breeding Research Station (CECA- Minas Gerais State, Brazil) who helped carrying out field trials and to collect phenotypic data.

## Author Contributions

**Conceptualization:** Mateus Teles Vital Gonçalves, Luiz Alexandre Peternelli.

**Data curation:** Mateus Teles Vital Gonçalves, Paulo Mafra de Almeida Costa, Pedro Marcus Pereira Vidigal, Luiz Alexandre Peternelli.

**Formal analysis:** Mateus Teles Vital Gonçalves, Pedro Marcus Pereira Vidigal, Luiz Alexandre Peternelli.

**Funding acquisition:** Marcio Henrique Pereira Barbosa, Luiz Alexandre Peternelli.

**Investigation:** Mateus Teles Vital Gonçalves, Paulo Mafra de Almeida Costa.

**Methodology:** Mateus Teles Vital Gonçalves, Gota Morota, Paulo Mafra de Almeida Costa, Pedro Marcus Pereira Vidigal, Luiz Alexandre Peternelli.

**Project administration:** Marcio Henrique Pereira Barbosa, Luiz Alexandre Peternelli.

**Software:** Luiz Alexandre Peternelli.

**Supervision:** Marcio Henrique Pereira Barbosa, Luiz Alexandre Peternelli.

**Visualization:** Mateus Teles Vital Gonçalves, Luiz Alexandre Peternelli.

**Writing – original draft:** Mateus Teles Vital Gonçalves, Pedro Marcus Pereira Vidigal.

**Writing – review & editing:** Gota Morota, Paulo Mafra de Almeida Costa, Luiz Alexandre Peternelli.

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
