## [Decision Letter · Decision Letter 0]

10 Aug 2020

PONE-D-20-21708

Near-infrared spectroscopy outperforms genomic selection for predicting sugarcane feedstock quality traits

PLOS ONE

Dear Dr. peternelli,

Thank you for submitting your manuscript to PLOS ONE. After careful consideration, we feel that it has merit but does not fully meet PLOS ONE’s publication criteria as it currently stands. Therefore, we invite you to submit a revised version of the manuscript that addresses the points raised during the review process.

We look forward to receiving your revised manuscript.

Kind regards,

Paulo Eduardo Teodoro, Dr.

Academic Editor

PLOS ONE

Additional Editor Comments:

Dear authors, your manuscript has been reviewed by two specialists. Both pointed out the need for Major Revision. Respond point-by-point to each request sent by reviewers.

Additionally, I send below some suggestions for your consideration:

- provide the keywords for your study;

- In the Introduction I missed explaining how the qualitative traits of sugarcane can be improved from the tested approaches. Consider providing more details about each of these traits;

- I also feel the need to explain better how NIR can be used for this culture. Are there any previous studies? This needs to be linked to the previous request in the Introduction;

- Apply the Tukey test to compare the strategies (M1, M2 and M3) presented in Figures 4 and 5.

Journal Requirements:

Reviewers' comments:

Reviewer's Responses to Questions

**Comments to the Author**

1. Is the manuscript technically sound, and do the data support the conclusions?

Reviewer #1: Yes

Reviewer #2: Partly

2. Has the statistical analysis been performed appropriately and rigorously? 

Reviewer #1: No

Reviewer #2: No

3. Have the authors made all data underlying the findings in their manuscript fully available?

Reviewer #1: No

Reviewer #2: Yes

4. Is the manuscript presented in an intelligible fashion and written in standard English?

Reviewer #1: Yes

Reviewer #2: Yes

5. Review Comments to the Author

Reviewer #1: The paper considers prediction of feedstock quality traits in sugar cane using genetic markers (GS) and NIRS. The results show that NIRS outperforms GS and that the combination of NIRS and genetic markers leads to no improvement over use of NIRS alone.

(1) Perhaps many readers will consider the result unsurprising because NIRS is much closer to the targeted traits than genetic markers. There are many similar reports in other crops and also with other high-throughput phenotyping data. A more thorough review of this growing literature would be useful.

(2) As the authors state in their introduction, the greatest potential of GS lies in the facility to predict traits even without phenotyping, thus permitting substantial shortening of breeding cycles. By contrast, the samples used for generating the NIRS data are obtained only 26 months after transplanting! Thus, with NIRS you are 26 months slower than GS! Wouldn't most breeders gladly take GS predictions now, even if they are a little less accurate, than waiting another 26 months before NIRS data are in?!

(3) The authors use an augmented design to test the 385 clones. They use a two-stage approach, where in the first stage they fit a linear mixed model with random effects for blocks and genotypes and then obtain BLUP of genotype effects to be carried forward to the second state (GS, NIRS-based prediction). There are three issues with this approach that require attention:

(i) It is not clear how the replicated checks enter the analysis. If a random effect for genotype is fitted, then it must be assumed that the checks come from the same population as the 385 clones. Is this a realistic assumption?

(ii) BLUP are shrinkage estimators. It has been argued by many several authors that these BLUP would need to be unshrunk before submission to the second stage. Animal breeders call these unshrunken estimators deregressed proofs. In animal breeding, use of BLUP is a necessity, essentially because bulls do not give milk and we can only assess their breeding value based on the performance of close female relatives. This is not an issue in plant breeding, where it is perfectly feasible to compute BLUE in the first stage. That this is a prefered approach, is explained, e.g., in Smith, A., B.R. Cullis, and A. Gilmour. 2001. The analysis of crop variety evaluation data in Australia. Aust. N. Z. J. Stat. 43:129–145.

(iii) The BLUE (or BLUP) computed from the augmented design in the first stage are not independent, nor do they usually have constant variance. It is therefore desirable to consider some form of weighting in the second stage. On this point, see Smith et al. (2001) and also Damesa et al. (2019): Comparison of weighted and unweighted stage-wise analysis for genome-wide association studies and genomic selection. Crop Science 59, 2572-2584.

(4) The authors use "random sampling cross-validation" (L247-248). Why not use k-fold cross-validation, as is much more common for these kinds of analysis?

(5) The authors correlate univariate BLUPs of the two target traits. What does this really estimate? I think it would be much better to fit a bivariate model and obtain a REML estimate of the genetic correlation.

(6) The results report a "prediction accuracy" (L277) but I think this criterion was never explained in M&M.

(7) A more gender-neutral translation of UAV is "uncrewed aerial vehicle", rather than "unmanned aerial vehicle". I am not insisting on this point, but am sure a lot of readers would appreciate this usage.

Reviewer #2: The manuscript addresses an important topic regarding the incorporation of new tools in plant breeding, aiming to improve accuracy, response to selection, and reduce time and costs.

However, there are two main issues to be solved:

The authors concluded that NIR outperforms GS. Regarding accuracy, it can be real. However, it is not a fair and complete comparison between these tolls. The former does not allow to reduce the cycle between generations. On the other hand, via GS, it is possible to speed up the breeding process. Therefore, I suggest include comparisons in terms of Response to Selection, dividing the accuracy by time (of each method).

Another concern is the GS model. The authors did not consider the dosage for each locus. Thus, all classes of heterozygous individuals are sum up in just one, reducing the resolution od the GS and estimation of the models. Furthermore, they did not consider dominance to predict hybrid clones, where heterosis is significant. I suggest both corrections.

6. PLOS authors have the option to publish the peer review history of their article (what does this mean?). If published, this will include your full peer review and any attached files.

Reviewer #1: No

Reviewer #2: No

---

## [Author Response · Author response to Decision Letter 0]

17 Jan 2021

We resubmitted our paper according to the considerations pointed out by the editor and reviewers. We have attached i) a rebuttal letter to the reviewers; ii) a marked-up copy of your manuscript that highlights changes made to the original version; and iii) an unmarked version of your revised paper.

---

## [Editor Report · Decision Letter 1]

21 Jan 2021

Near-infrared spectroscopy outperforms genomic selection for predicting sugarcane feedstock quality traits

PONE-D-20-21708R1

Dear Dr. peternelli,

We’re pleased to inform you that your manuscript has been judged scientifically suitable for publication and will be formally accepted for publication once it meets all outstanding technical requirements.

Kind regards,

Paulo Eduardo Teodoro, Dr.

Academic Editor

PLOS ONE

Additional Editor Comments (optional):

The authors carefully reviewed the manuscript according to my suggestions and that of the reviewers. Therefore, I recommend that the manuscript be accepted for publication in Plos One.

---

## [Editor Report · Acceptance letter]

8 Feb 2021

PONE-D-20-21708R1 

Near-infrared spectroscopy outperforms genomics for predicting sugarcane feedstock quality traits 

Dear Dr. peternelli:

I'm pleased to inform you that your manuscript has been deemed suitable for publication in PLOS ONE. Congratulations! Your manuscript is now with our production department. 

Kind regards, 

on behalf of

Professor Paulo Eduardo Teodoro 

Academic Editor

PLOS ONE